# A scoping review of the landscape of health-related open datasets in Latin America

**David Restrepo**[1,2]*, **Justin Quion**[1], **Constanza Vásquez-Venegas**[3], **Cleva Villanueva**[4], **Leo Anthony Celi**[1], **Luis Filipe Nakayama**[1,5]

1 Laboratory for Computational Physiology, Massachusetts Institute of Technology, Cambridge, Massachusetts, United States of America, 2 Telematics Department, University of Cauca, Popayán, Cauca, Colombia, 3 Scientific Image Analysis Lab, Integrative Biology Program, Biomedical Sciences Institute (ICBM), Faculty of Medicine, Universidad de Chile, Santiago, Chile, 4 Instituto Politécnico Nacional, Escuela Superior de Medicina, Ciudad de Mexico, Mexico, 5 Department of Ophthalmology, São Paulo Federal University, São Paulo, São Paulo, Brazil

* davidres@mit.edu

## Abstract

Artificial intelligence (AI) algorithms have the potential to revolutionize healthcare, but their successful translation into clinical practice has been limited. One crucial factor is the data used to train these algorithms, which must be representative of the population. However, most healthcare databases are derived from high-income countries, leading to non-representative models and potentially exacerbating health inequities. This review focuses on the landscape of health-related open datasets in Latin America, aiming to identify existing datasets, examine data-sharing frameworks, techniques, platforms, and formats, and identify best practices in Latin America. The review found 61 datasets from 23 countries, with the DATASUS dataset from Brazil contributing to the majority of articles. The analysis revealed a dearth of datasets created by the authors themselves, indicating a reliance on existing open datasets. The findings underscore the importance of promoting open data in Latin America. We provide recommendations for enhancing data sharing in the region.

## Author summary

In this review, we explore the potential of artificial intelligence (AI) algorithms to revolutionize healthcare while addressing the challenges of translating them into clinical practice. One crucial obstacle we identify is the limited availability of representative data to train these algorithms. Most healthcare databases are sourced from high-income countries, resulting in non-representative models that may worsen health inequities. Our focus is on health-related open datasets in Latin America, where we aim to identify existing datasets, analyze data-sharing frameworks, techniques, platforms, and formats, and highlight best practices in the region. Through our analysis, we found 61 datasets from 23 countries, with the majority relying heavily on the DATASUS dataset from Brazil. Surprisingly, there is a lack of datasets created by the authors themselves, indicating a reliance on existing open datasets. Our findings underscore the urgent need to promote open data initiatives in Latin America, and we provide recommendations for enhancing data sharing

**Data Availability Statement:** The retrieved articles, reviewers assessment, and codes are publicly available at: https://github.com/dsrestrepo/MIT_Review_datasets_Latin_America.

**Funding:** The authors received no specific funding for this work.

in the region. By fostering data accessibility, we can unlock the potential of AI to advance healthcare for all.

## Introduction

Artificial intelligence (AI) algorithms hold great promise in healthcare, enhancing clinical decision-making, diagnosis, and identifying new genomic patterns and drugs [1,2]. However, few AI systems have been translated into clinical practice, and those that have been have not shown much success [3–5].

Much of the machine learning community around the world has focused on generating new algorithms and more complex machine learning techniques like Transformer models such as BERT [6], GPT [7] or Stable Diffusion [8]. However, there is a lack of research investigating the data utilized by these algorithms and whether or not it is representative of the population in question. Algorithms that are trained on non-representative data can lead to dangerous and biased outcomes [9,10]. More often than not, it is those who are underrepresented in the data who will be harmed the most, such as those in low and middle-income countries (LMICs), women [11], and non-Whites [12]. If left unchecked, these encoded biases will continue to generate inequities in health and widen the gap between populations [13].

An important component of data, and generally overlooked, is the amount, quality, and accessibility of datasets. As it stands now, most healthcare databases originate from high-income countries [14–16], limiting the generalizability of any resulting models. Quality is another important aspect of datasets and one that is difficult to come by especially in healthcare due to the personal nature of the information. This data must be deidentified which can be a laborious and expensive process. Finally, a dataset has little value if it cannot be accessed [17]. Publicly available datasets promote reproducibility, enable validation studies, and are a valuable alternative to the elevated costs and challenges of developing a database [14,18].

This review seeks to explore the landscape of health-related open datasets in Latin America. Specifically, the authors aim to identify the existing open health datasets in Latin America through a mapping of the existing literature in Latin American countries; take note of the modalities, techniques, platforms and formats being used to share data; and highlight the initiatives and practices around the publication of open data in Latin America. Additionally, limitations and gaps surrounding the current landscape of health data sharing in Latin America will be identified. Finally, recommendations and suggestions to promote the use of open data in Latin America are provided.

## Results

The initial search yielded 700 papers (Fig 1) and 170 duplicates were removed. From the 530 papers that remained, a primary screening based on title and abstract assessment excluded 344 papers. Finally a full-text analysis was performed and excluded 45 papers. A total of 141 documents were used in the final quantitative and qualitative analysis.

The 141 remaining articles were published between 2006 and 2023, with the majority 96 (68.1%) originating from 2020 to 2022.

### Authors

In 120 (85.1%) of the articles, at least one author was working in a Latin American institution. It should be noted nationality was based on the location of the affiliated institution which

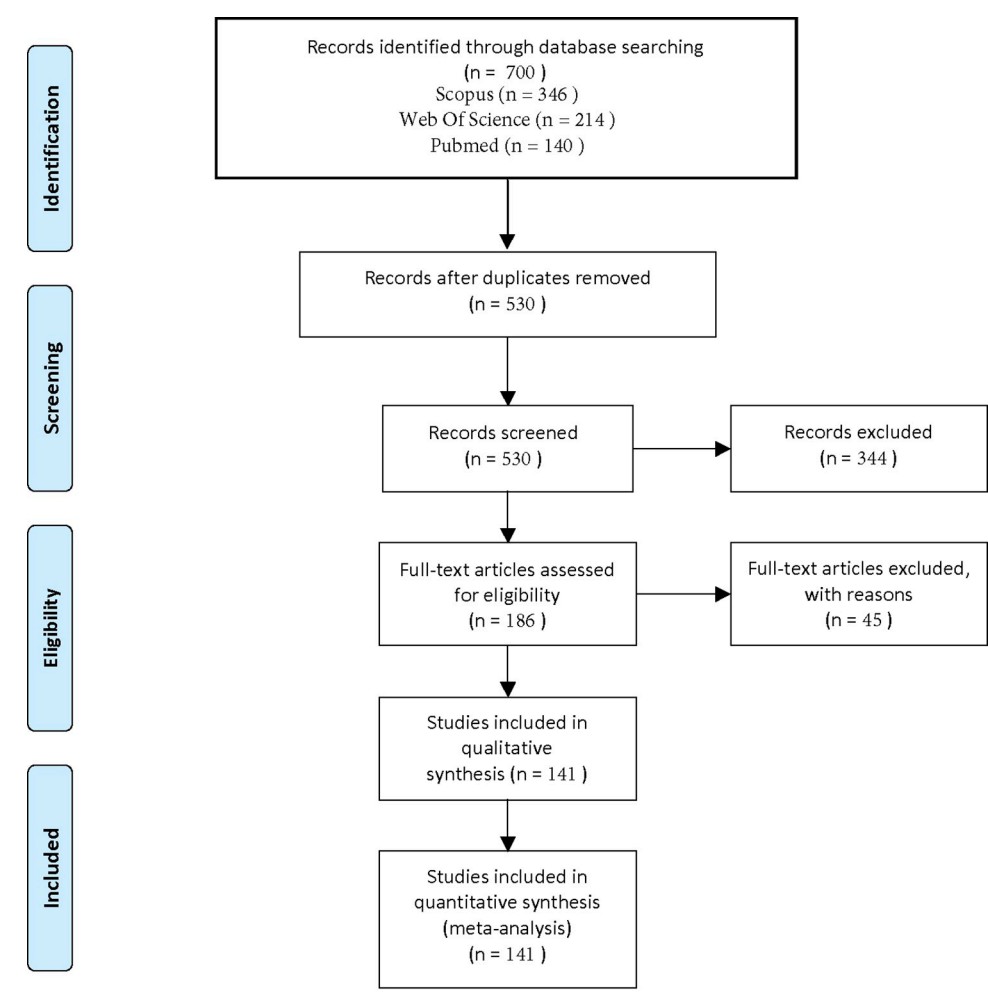

*From:* Moher D, Liberati A, Tetzlaff J, Altman DG, The PRISMA Group (2009). *Preferred Reporting Items for Systematic Reviews and Meta-Analyses: The PRISMA Statement.* PLoS Med 6(7): e1000097. doi:10.1371/journal.pmed1000097

**Fig 1. Flow diagram for article inclusion and exclusion from PRISMA.**

ignores the possibility of Latin American authors immigrating and working at other institutions and vice-versa. Current infrastructure does not allow for easy parsing of author nationality and is outside the scope of this review.

## Datasets

This review identified 61 datasets (38 described in Table 1 and 23 described in Table 2) from 23 countries. From those identified datasets, the contribution of the Brazilian database DATA-SUS dataset stands out, which contributed to 54 (38.3%) of the 141 documents. The contribution of the Brazilian Institute of Geography and Statistics (IBGE) also stands out as the main source of social determinants of health in Brazil, contributing to 9 articles (6.4%). Finally, datasets available for all of Latin America contributed to 43 articles (30.5%). It is important to note

**Table 1. Open databases found in Latin American countries.** The databases resulting from articles that created the database and released it are not mentioned here because they will be mentioned later. All Latin America means Argentina, Bolivia, Brazil, Chile, Colombia, Costa Rica, Cuba, Dominican Republic, Ecuador, El Salvador, Guatemala, Guyana, Haiti, Honduras, Mexico, Nicaragua, Panama, Paraguay, Peru, Puerto Rico, Suriname, Uruguay, Venezuela.

| Name of Dataset | Country of Dataset | Description | Number of Papers Using The dataset |
|---|---|---|---|
| Brazilian Public Health System Database (DATASUS) | Brazil | Datasets provided by the Informatic Brazilian Department of Unified Health System, established in 1991. | 54 |
| Brazilian Institute of Geography and Statistics (IBGE) | Brazil | Datasets provided by the Brazilian Institute of Geography and Statistics. IBGE provide census and healthcare-related data. | 9 |
| CRESI-SINTRA (Argentinian Database of the National Procurement Organization) | Argentina | The CRESI-SINTRA database is an open dataset maintained by the National Procurement Organization in Argentina, providing information on liver transplant recipients. | 1 |
| John Hopkins COVID-19 Dashboard | All Latin America | Database of COVID-19 cases, deaths, recoveries, testing, collected from multiple data sources. | 5 |
| World Bank | All Latin America | The World Bank's COVID-19 (Coronavirus) Response dataset. Global pandemic's impact on various sectors and regions worldwide. | 3 |
| Open data on COVID-19 from the Ministry of Health of each country | All Latin America | Each Latin American country provides an open COVID-19 dataset with information on the cases, deaths, and other relevant statistics of COVID-19. | 15 |
| Audifarma SA Pharmacovigilance ADR Reports | Colombia | The pharmacovigilance dataset of Audifarma SA contains reports of adverse drug reactions (ADRs). | 1 |
| Global Burden of Disease Study 2019 (GBD 2019) | All Latin America | Available at ghdx.healthdata.org. Is a dataset that quantifies the impact of various diseases, injuries, and risk factors on human health and mortality | 2 |
| Food and Agriculture Organization Corporate Statistical Database (FAOSTAT) (https://www.fao.org/faostat/en/#home). | All Latin America | FAOSTAT is the Food and Agriculture Organization's official corporate statistical database, offering a collection of global agricultural, food, and environmental data. | 3 |
| Brazilian Institute for Applied Economic Research (IPEA) (http://www.ipeadata.gov.br/) | Brazil | The Brazilian Institute for Applied Economic Research (IPEA) maintains a web portal with a database of economic indicators and socio-economic data for Brazil. | 2 |
| Open data from the Ministry of Health of each country | All Latin America | Some documents do not specify the name of the dataset, however they mention having obtained the data from the Ministries of Health of their respective country. | 4 |
| System of Information of Social Protection (SISPRO) | Colombia | SISPRO, provides information that cover various aspects such as healthcare service availability, service quality, insurance coverage, financing, and social promotion. | 1 |
| Twitter API | All Latin America | Is a programming interface provided by Twitter that allows access and interaction with Twitter's platform, enabling it to retrieve tweets. | 1 |
| Endoscopic ultrasound database of the pancreas. http://cimalab.unal.edu.co/?lang=es&mod=program&id=26 | Colombia | The Endoscopic Ultrasound Database of the Pancreas is a specialized dataset hosted by the National University of Colombia, featuring comprehensive information on pancreatic conditions obtained through endoscopic ultrasound procedures. | 1 |
| Spiral HandPD and NewHandPD from Botucatu Medial School, Sao Paulo State University, Brazil | Brazil | Spiral HandPD and NewHandPD are datasets developed by the Botucatu Medical School at São Paulo State University in Brazil. These datasets focus on hand movement data and aim to aid in the study and diagnosis of Parkinson's disease. | 1 |
| Databases of the Contributory Regime of the Social Security System in Health of Colombia | Colombia | The databases of the Contributory Regime of the Social Security System in Health of Colombia comprise comprehensive records of individuals' contributions and coverage within the country's healthcare system. | 1 |
| WHO mortality database | All Latin America | The WHO mortality database contains data on cancer-related deaths, providing valuable insights into global mortality rates and trends associated with various types of cancer. | 2 |
| WHO FluNet | All Latin America | s an open dataset that provides global surveillance data on influenza, including information on its prevalence, strains, and geographic distribution. | 1 |
| Pan American Health Organization. Regional Core Health Data Initiative. | All Latin America | Hosted by the Pan American Health Organization (PAHO), is an effort aimed at collecting and sharing essential health data across the Americas region. | 1 |

(*Continued*)

**Table 1.** (Continued)

| Name of Dataset | Country of Dataset | Description | Number of Papers Using The dataset |
|---|---|---|---|
| ELSI-BRAZI | Brazil | Brazilian Longitudinal Study of Aging, an extensive dataset focused on studying the aging process, health, and socioeconomic aspects of the Brazilian population. | 1 |
| SIASG | Brazil | It is a Brazilian government system that manages and controls public procurement and contracting processes. | 1 |
| SINADEF | Peru | A database that records and tracks mortality data in Peru, including causes of death, demographics, and other relevant information. | 1 |
| Our world in data | All Latin America | Dataset that offers extensive information and statistics on the COVID-19 pandemic | 1 |
| COVID-19 Albert Einstein Hospital | Brazil | The COVID-19 dataset from Albert Einstein Hospital includes medical data and records of patients to which the PCR test was applied for the detection of covid | 1 |
| ENAHO | Peru | Is a dataset in Peru that collects information on various socio-economic aspects such as income, education, employment, housing, and health | 2 |
| DGIS | Mexico | Information of health in Mexico | 4 |
| INEGI | Mexico | Health and socio-demographic open data set from Mexico. Information is provided as a country and divided in states and municipalities | 4 |
| Guatemala RCT Database (PhysioNet.org) | Guatemala | The dataset comprises diverse data related to physiological measurements, clinical outcomes, and other relevant variables. It aims to support research and analysis in the field of healthcare by providing valuable insights into the effects of various interventions, treatments, or experimental conditions on the population under study | 1 |
| Puerto Rico Heart Health Program (PRHHP) | Puerto Rico | This dataset is designed to capture a wide range of information pertaining to cardiovascular health, including risk factors, medical history, lifestyle habits, and diagnostic measures. It aims to provide valuable insights into the prevalence, determinants, and outcomes of heart-related conditions among individuals in Puerto Rico | 1 |
| Google Maps (GM) | All Latin America | It provides a vast dataset of geospatial information, including maps, satellite imagery, and street views from around the world. GM collects and stores data on geographical features, landmarks, points of interest, transportation networks, and more | 1 |
| National Survey of Social Exclusion (NESS) | Mexico | This dataset provides valuable information about various aspects of life in Mexico, including income distribution, education levels, employment rates, access to basic services, housing conditions, and other socio-economic indicators | 1 |
| WHO Global Health Estimates | All Latin America | It encompasses a wide range of health-related indicators and estimates, providing insights into the global burden of disease, mortality rates, life expectancy, risk factors, and health system performance. | 1 |
| 2018 Mexican Social Security Institute dataset | Mexico | A dataset pertaining to social security in Mexico. It encompasses various aspects of the institute's operations, including information on healthcare services, pensions, employment, and contributions | 1 |
| CPC (Carolina Population Center) | All Latin America | Is a collection of demographic and population-related data compiled and maintained by the Carolina Population Center, a research institute based at the University of North Carolina at Chapel Hill. | 1 |
| SISVER (Sistema de Vigilancia Epidemiológica de las Rabias) | Mexico | This dataset collects and manages information related to rabies cases, including animal bites, human cases, animal species involved, geographic location, and other relevant epidemiological factors. | 1 |
| World Meters | All Latin America | Provides real-time statistics and data on various global metrics such as population, health, education, economy, and more | 1 |
| Mexican-Mestizo patient transcriptome dataset (GSE56303) | Mexico | available on the Gene Expression Omnibus (GEO) database. Is collection of genetic information specifically obtained from individuals of Mexican-Mestizo descent. This dataset includes transcriptomic data, which encompasses the genetic activity and expression levels of various genes across different tissues or cell types within these patients. | 1 |

(*Continued*)

**Table 1.** (Continued)

| Name of Dataset | Country of Dataset | Description | Number of Papers Using The dataset |
|---|---|---|---|
| Genome Aggregation Database (gnomAD) | All Latin America | dataset that focuses on aggregating genetic information from diverse human populations. It contains genomic data derived from thousands of individuals, encompassing both exome sequencing and whole-genome sequencing data. | 1 |

that many of the datasets included are global in scope and are not specific to Latin America alone.

## Country

Brazil is the most prevalent country, mainly due to the influence of DATASUS, appearing in 83 (58.7%) of the papers, followed by Mexico with 47 (33.3%) documents, and Colombia with 32 (22.7%). Guyana and Suriname, on the other hand, are the countries that are present in the least number of articles, appearing only 14 (9.9%) and 13 (9.2%) times, respectively. A heatmap showing the distribution of paper appearances between countries is seen in Fig 2.

## Local datasets

Despite the large number of articles resulting from the search, articles rarely generated and analyzed datasets from the authors' institutions. Instead, many of the datasets used were generated by governments and NGOs. Of the 141 resulting papers, only 23 of them created their own dataset. The other 118 papers utilized public datasets.

From those papers that created open datasets, 12 of those datasets were created for specific Latin American countries. 5 datasets were created for Colombia: 1 with social determinants of health and nutrition data for Public Health tasks [21], and 4 with a more clinical point of view which are a muscle dysmorphia dataset [22], body fat measurement [23], endoscopic ultrasound scans [24], and a treatment of Helicobacter pylori dataset [25]. 8 datasets were created for Brazil: 2 from a Public Health perspective which are BASICS [26] with epidemiologic data, and the dataset on child vaccination [27]; 1 dataset of laboratory exams [28]; 1 dataset of images of leprosy called AI4Leprosy [29]; 1 open dataset with electronic health records called ORBDA [30]; 2 datasets for genomics [31,32]; and one trial with the effects of BCG vaccination for COVID-19 to [33]. Of note, 4 datasets were created for Mexico, followed by Cuba (1) and Honduras (1). The rest of the datasets were created globally, mainly for public health, and included either some or all Latin American countries. The full list of datasets and more information about the datasets can be seen in Table 2.

## Open vs. credentialed users

Of the total datasets in Latin America, only 7 (5%) require credentialing. Three of these correspond to databases created in research papers (Table 2): SELAdb database [31,32], the Helicobacter pylori dataset [25], and the Cuban Human Brain Mapping Project (CHBMP) [42].

## Dates

A majority of the papers were published from 2020–2022 (Fig 3), a spike that may be fueled by the COVID-19 pandemic. Of the 96 papers published during that time period, 33 (34.4%) focused on COVID, either in Latin America or utilized Latin America data.

**Table 2. Open datasets in Latin America generated through articles found in the review.** Although only Latin American countries appear in the table in the column "Latin American Countries", it is also important to note that in many cases those are global datasets that are also available for countries in other continents. All Latin America means Argentina, Bolivia, Brazil, Chile, Colombia, Costa Rica, Cuba, Dominican Republic, Ecuador, El Salvador, Guatemala, Guyana, Haiti, Honduras, Mexico, Nicaragua, Panama, Paraguay, Peru, Puerto Rico, Suriname, Uruguay, Venezuela.

| Dataset Name | Latin American Countries | Description | Paper |
|---|---|---|---|
| BCG Vaccination to Protect Healthcare Workers Against COVID-19 (BRACE) | Brazil | Random trial dataset to assess the effectiveness of BCG vaccination in symptomatic and severe COVID-19. The dataset contains data of 10,078 healthcare workers from Australia, The Netherlands, Spain, the UK, and Brazil. | [33] |
| BASICS—Spatial, demographic, and socioeconomic data for epidemiologic research | Brazil | A dataset with information of various open sources data Brazil. The dataset contains 139,153 rows and 26 columns with socioeconomic, sociodemographic and epidemiological information. | [26] |
| Latin America and the Caribbean (LAC) population dataset | All Latin America | Dataset that provides birth and pregnancy information in maps of African, Latin American, and Caribbean countries. | [34] |
| Dataset for estimation of muscle dysmorphia | Colombia | A dataset with information of 200 individuals attending local gyms labeled as normal weight, overweight, or obesity according to WHO guidelines. The dataset provides information on physical activity, food supplement habits, psychological pressure, and risk of muscle dysmorphia. | [22] |
| Dataset for new body fat measurement. | Colombia | The dataset provides information on body fat measurement using bioelectrical impedance analysis. The dataset contains 345 patients aged between 18 and 60 years in Barranquilla, Colombia. | [23] |
| Dataset on child vaccination | Brazil | Dataset with information of children vaccination in Brazil. The dataset contains information of children under five years in Brazil from 1996 to 2021. The dataset includes 2,442,863 observations, 35 attributes, and 1,344,480,329 vaccine doses for 15 diseases. | [27] |
| Database of endoscopic ultrasound scans of the pancreas. | Colombia | The dataset contains endoscopy ultrasound videos with 55 cases with B-mode ultrasound images from two medical units in Colombia. The dataset includes 18 cases of pancreatic cancer, 5 cases of pancreatitis, and 32 cases of healthy pancreas, liver, and gallbladder. | [24] |
| WordPop-RF | All Latin America | A high-resolution dataset containing information of the population distribution in the Latin American countries in 2010, 2015 and 2020 stored in a map. | [35] |
| Laboratory exams of the National Health Survey. | Brazil | A dataset with laboratory exams from 8,952 individuals in Brazil selected randomly from the census. The dataset includes blood and urine tests. | [28] |
| Multidimensional Dataset Of Food Security And Nutrition In Cauca | Colombia | A Multidimensional Dataset for characterizing food security in Colombia. The dataset includes 926 attributes incorporating various sources such as population and agricultural census, nutrition surveys, and satellite imagery. | [21] |
| ORBDA | Brazil | openEHR Benchmark Dataset (ORBDA) is a large healthcare benchmark dataset encoded in the openEHR format. It is derived from a de-identified dataset from the Brazilian National Healthcare System and includes hospitalization and high complexity procedures information. The dataset, available in XML and JSON formats, consists of over 150 million composition records | [30] |
| AI4Leprosy | Brazil | Combines skin images and clinical data to aid in the diagnosis of leprosy. The dataset used includes 1229 skin images and 585 sets of metadata collected from 222 patients with leprosy or other dermatological conditions | [29] |
| World Mortality Dataset | All Latin America except Venezuela, Guyana, Suriname and Honduras. | A regularly updated World Mortality Dataset that collects all-cause mortality data from 103 countries and territories. | [36] |
| Treatment regimens used in the management of Helicobacter pylori in Colombia | Colombia | Medication dispensing database in Colombia to identify the regimens used for Helicobacter pylori eradication over a 6-year period. | [25] |
| Exome Evaluation of Autism-Associated Genes in Amazon American Populations | Brazil | A database that includes genetic variants of CHD8, SCN2A, FOXP1, and SYNGAP1 genes in the Amazonian Amerindian population. | [31,32] |
| SELAdb | Brazil | A genomic database that includes a cohort of 523 unrelated individuals, mainly of mixed Latin American origin. | [31,32] |

*(Continued)*

**Table 2.** (Continued)

| Dataset Name | Latin American Countries | Description | Paper |
|---|---|---|---|
| Medicines Quality Database (MQDB) | Bolivia, Guyana, Peru, Colombia, Ecuador | A medicines quality database (MQDB) containing data from medicines quality monitoring (MQM) activities in 17 countries across Africa, Asia, and South America from 2003 to 2013. Includes information on 15,063 samples, primarily antibiotics, antimalarials, and antituberculosis medicines, tested using Minilab | [37] |
| A behavioral, clinical and brain imaging dataset with focus on emotion regulation of females with fibromyalgia | Mexico | The dataset contains MIR data of 66 female patients, 33 with fibromyalgia and 33 healthy controls. | [38] |
| SUPEREGO urinary metabolomics | Mexico | A dataset of military personnel with urinary metabolomics liquid-chromatography coupled to mass spectrometry data. The dataset contains information of patients with normal weight, overweight, or obese. | [39] |
| CSS Mexico | Mexico | A dataset that contains the results of a survey to measure the level of stress due to COVID-19 in Mexico during July and August 2020 | [40] |
| EPIC2 | Honduras | The dataset contains all the information necessary for the EPIC studies. The data includes general facility information, immunization-related facility information, inputs (labor, vehicles, buildings), outputs (dose, FIC, DTP and measles), total cost and cost per input and activity. | [41] |
| Cuban Human Brain Mapping Project (CHBMP) | Cuba | Is a dataset focused on neuroimaging research conducted in Cuba. This dataset encompasses a wide range of data related to the human brain, including neuroimaging scans, cognitive assessments, and demographic information. | [42] |
| SUDMEX CONN | Mexico | A dataset that contains demographic, cognitive, clinical, and magnetic resonance imaging (MRI) data from 120 patients with Cocaine use disorder (CUD) in Mexico. The dataset contains data of 74 CUD patients and 64 healthy controls | [43] |

## Data type

It is also important to take into account the types of data, since a wide range of sources and forms of data also enable a wide range of types of models and investigations in the territory. Fig 4 shows the distribution of data types in the found databases. The most utilized data type is tabular data used in 125 (88.7%) of the papers, followed by images with 5 (6.4%) and finally genomic data, signals, and text with 4 (6.4%), 2 (1.4%) and 1 (0.7%) respectively. Out of the 23 datasets that were created by papers, 15 (65.2%) were tabular, 6 (26.1%) used images and 2 (8.6%) were genomic data. Text datasets were not presented in the papers that generated their own datasets. It should be noted that some papers used multiple datasets and combined different forms of data. Additionally, DATASUS is primarily a tabular dataset and its prevalence throughout the papers skews the results.

## Discussion

The landscape of publicly-available datasets in Latin America is still in its infancy. While Brazil has made great strides around open data mainly through its DATASUS platform, the other Latin American countries are lagging behind. This is due to two reasons, either the current infrastructure—personnel and systems—does not support the creation of datasets in the organizations and they simply do not exist, or despite the existence of health datasets in these countries, they are not being curated and analyzed for a variety of reasons ranging from awareness to accessibility.

Based on the results, there is much to be improved upon. A majority of the papers and datasets used originate from Brazil and many Latin American countries are not represented.

## Papers product of open datasets in Latin America

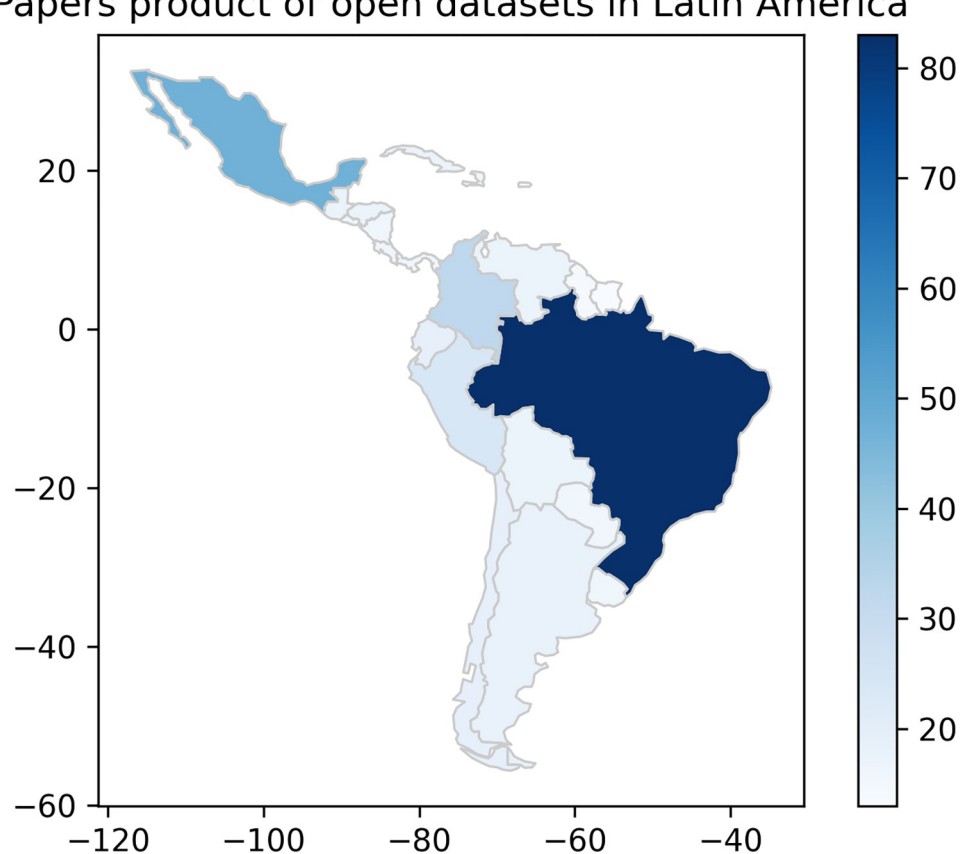

**Fig 2. Heat map showing the distribution of articles published using open data in Latin America.** The most intense colors indicate a greater presence of articles and the lighter colors indicate less presence. Map created adapting naturalearth_lowres's layer using (c) 2013–2022, GeoPandas developers, an open source python package created under the liberal terms of the BSD-3-Clause license [49].

Despite the robustness of the DATASUS databases, models developed solely with this data will have limited applicability outside of Brazil. As Movva et al have shown [9], the use of broad demographic groups can hide disparities in the subpopulations. In our case, Brazil alone is not representative of Latin America. South and Central American countries have specific sociocultural and other factors that are unique from each other.

Even when the other Latin American countries are represented in the data, on many occasions it is not due to local initiatives, but rather by initiatives of international organizations that collect and release this data globally. It is not sufficient to simply be represented by data alone; it is crucial that the communities investigating these problems are also representative of the population in question. These groups will be more attuned to the unique socio-cultural context of the problem and are more likely to come up with solutions.

The number of publicly available healthcare datasets originating from Latin America is limited, with DATASUS being the primary database used and providing limited options for research questions to be addressed. The problem becomes more evident given that of the existing datasets, 88.7% aggregate data at the level of population living in cities or even countries. In terms of topics and modalities, the search did not find any clinical dataset. Of the datasets found, only 6.4% were of medical images, making research in specialties such as radiology and ophthalmology not possible.

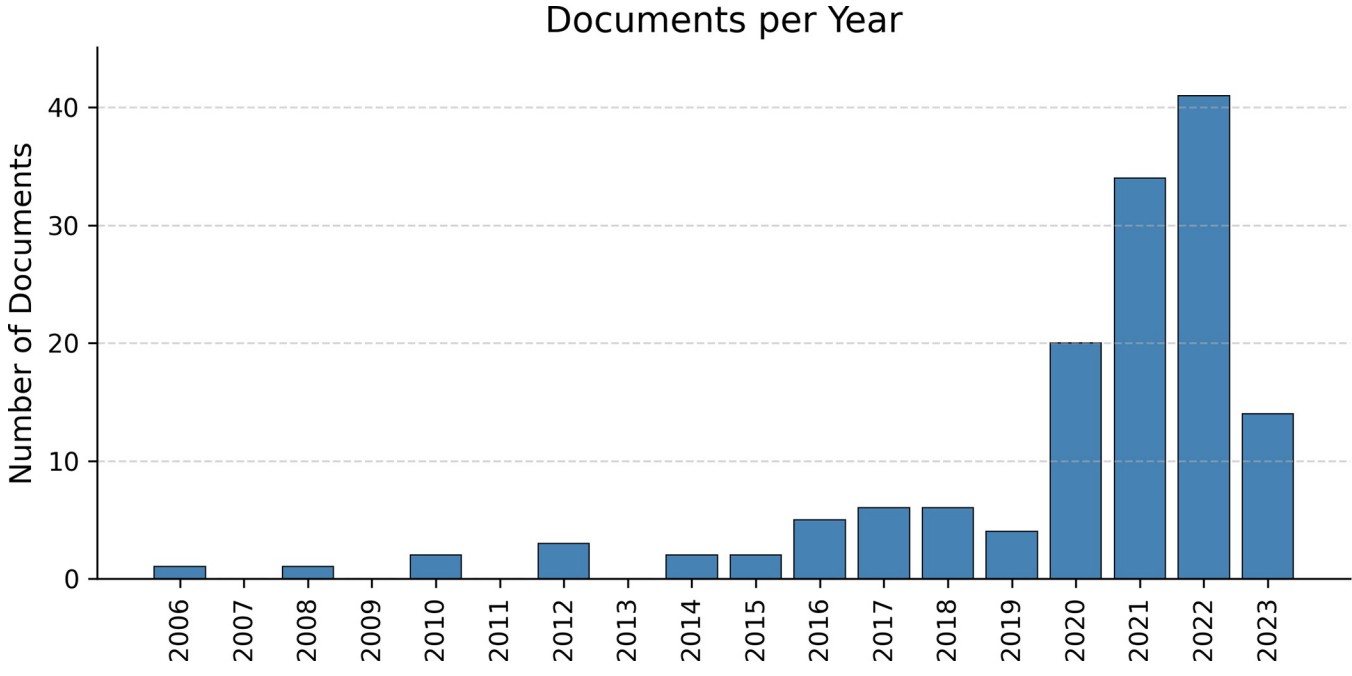

**Fig 3. Number of papers published that created open datasets in Latin America or that used open datasets created in Latin America.**

It is crucial that open data becomes more mainstream in order to promote transparent and reproducible health research [17,44], support processes and quality improvements in health systems, and mitigate algorithmic biases [45,46] as the interest in artificial intelligence intensifies. Sharing data should be at the core of the scientific process to ensure reproducibility especially in the area of health where lives of patients are at stake.

Compliance with FAIR (findable, accessible, interoperable, and reusable) [47] practices for health data sharing in Latin American articles falls short, generating a lack of reproducibility, research advancement, and reduction of health inequities. Prioritizing the adherence to the FAIR principles should be crucial for datasets in Latin America. It is important to bear in mind that the existence of data is not enough if there are many barriers involved. Simply having the data is not enough; it must be available and in optimal conditions for effective utilization.

This review has limitations. The data sources for extracting the articles were limited to 3 databases (Scopus, Web Of Science, and PubMed), and despite the combined scope, some papers and databases may have been omitted, although we argue that databases should be "findable" if they are to be useful. It should also be taken into account that many of the data sources found do not come from scientific publications, but are data sources published by governments and/or local or international organizations. We understand that many of the open data sources in Latin America may not be present in scientific databases and, therefore, may have been excluded from this review. Other data sources, such as open data repositories, should be explored in future reviews.

## Recommendations

### Support funding and infrastructure

It is necessary to support, encourage, and streamline the creation and maintenance of datasets through funding and infrastructure. Not only should governments and funding agencies assist

## Data Types Distribution

**Fig 4. Data Type used in datasets from Latin America in all documents.**

but other institutions such as hospital and research groups should be encouraged to partake in order to increase the robustness of the data and datasets available. This includes other forms of data as well such as images and text. Investments in technological infrastructure are also necessary as the storage and processing of the data are key components that cannot be overlooked.

While data sharing policies vary across the region, some countries are aligning with global practices by incorporating the obligation to open data release, often tied to funding support. These policies promote transparency and accessibility, highlighting the need for dedicated funding mechanisms and enhanced data infrastructure. These policies can foster data-sharing cultures and empower Latin America in the open data aspects.

### Improve data quality and standardization

Attention should be given to the quality and standardization of health datasets in Latin America in order promote and facilitate the use of open data. A good first step towards this goal would be the establishment of guidelines and protocols regarding collection, storage, and sharing of data. Other possibilities include establishing a rigorous de-identification process and implementing data governance practices. Practices, such as controlled access can also be used in cases where de-identification of medical data is not possible. In certain cases, using

credentialed data access, rather than fully open data, may be a suitable approach, ensuring that data security and ethical considerations are addressed.

Incorporating benchmarks is also extremely important in the context of open health data in Latin America. To facilitate this, we propose the use of open-source and free tools like GitHub or GitLab, together with programming languages like Python or R, to run benchmark models and ensure reproducibility. Benchmarks not only provide a standardized framework for evaluating the quality and performance of open health datasets but also serve as a foundation for ongoing research and collaboration. By establishing benchmarks, the Latin American research community can accelerate the development of research-based algorithms and encourage the generation of additional data, thus fostering innovation and collaboration.

### Foster data sharing culture

Encouraging a culture of data sharing is crucial for the advancement of open data initiatives in Latin America. Key steps to achieve this include recognition, academic or otherwise, and incentives for those who share data openly, such as funding opportunities. Health institutions should be made aware that sharing data is an important but delicate process. Incorporating data treatment policies to avoid risks related to data privacy and identification is vital in each institution. However, sharing data represents more benefits than risks for health institutions.

### Address ethical considerations

As the use of open data grows, it is important to consider the ethical ramifications of its use, ranging from data privacy to the potential biases within. Clear guidelines and frameworks should be established to ensure the responsible and ethical use of such data. Additionally, frequent investigations into the datasets may be necessary in order to secure future use of data and models.

### Promote multidisciplinary collaboration

As data becomes accessible, the various fields become more entwined and as a result, a multidisciplinary approach is necessary. Health disparities and inequities cannot be tackled by health researchers and data alone. Instead, local communities should be engaged at all levels to better identify solutions with the unique sociocultural perspective in mind.

## Conclusion

The authors have performed a review of the research carried out in Latin America that yielded 141 articles utilizing open data related to health without any time, citation or year limitations. From these 141 articles, data such as country of authors' affiliations, most utilized data sources, data type, and a description of these datasets was extracted. As a conclusion:

- Having a standardized and accessible data source, as is the case of DATASUS in Brazil, also generates a great source of resources for local research, decision-making, and development in the region.

- Latin America is a region in which work is still needed around open data from many perspectives such as governments and open data policies. This problem can be seen very easily, especially when comparing the number of articles generated by data from the local health ministries of each country with data generated through international organizations such as NGOs. Special emphasis must be placed on the fact that it is not just having the data, but

how to release and share it, since on many occasions the NGO datasets are derived from reports from the Ministries of Health of each region.

- It is recommended that in Latin America a multidimensional approach be taken where different stakeholders from the government, research institutions, health organizations, among others, work together to create an open health ecosystem. These kind of initiatives can be done through events such as datathons [48], conferences and congresses in which topics related to local needs are worked with local experts in multidisciplinary teams. It must be avoided that the rules and research in the region continue to be carried out by people with total ignorance of the problem.

Implementing these kinds of efforts requires adequate equipment, resources, and a long-term commitment from all parties. In any case, it must be taken into account that the possible benefits of these changes would be much greater in the long term for the improvement of health ecosystems, reduction of biases and inequities in health in Latin America.

## Materials and methods

This scoping review focused on publicly available Latin American datasets using a PRISMA methodology [19]. The literature review search included PubMed, Scopus, and Web of Science databases, before 21, June 2023. The search strategy utilized variations of keywords "dataset" and "publicly available", along with Latin American countries. Document types were limited to journal papers, conference papers, and data papers with no language, year, or citations exclusion. Exact search criteria can be found in S1 File.

Papers were considered eligible if they met the following criteria: i) Papers were published in academic journals, conference proceedings, and reputable sources; ii) Studies that focus on health-related open datasets in Latin America; iii) Studies that provide information on the availability, accessibility, and use of open health datasets; plus any of the following criteria: a) Studies that discuss the modalities, techniques, platforms, and formats used for sharing data in Latin America; b) Studies that highlight initiatives and practices related to the publication of open data in Latin America; c) Studies that identify limitations, gaps, and challenges in the current landscape of health data sharing in Latin America.

Papers were excluded if they did not meet all of the following: i) Non-academic sources such as blog posts, opinion pieces, and news articles; ii) Studies not focused on health-related datasets or not specific to Latin America; iii) Studies that do not discuss the availability, accessibility, or use of open datasets; iv) Studies that are not related to the modalities, techniques, platforms, or formats used for sharing data in Latin America; and v) Studies that do not address initiatives and practices related to the publication of open data in Latin America.

Analysis of documents and the assessment process for eligibility was performed by three authors (DR, LFN, and JQ) where each paper was reviewed at least 2 times by two of the authors to ensure eligibility. As a first step, the software ScientoPy [20] was utilized to clean and remove duplicates from the search results. Since ScientoPy only works for Scopus and Web Of Science, it was necessary to remove duplicates from Pubmed through an alternative method. Python 3.10.12 programming language was leveraged to check the DOIs from the preprocessing of Scopus and Web Of Science, and remove those that were also present in the Pubmed result. Finally, two screening steps were employed: first by title and abstract, then by full-text assessment. All authors participated in the full-text assessment.

The retrieved articles, reviewers assessment, and codes are publicly available at: https://github.com/dsrestrepo/MIT_Review_datasets_Latin_America

### Reviewed variables

From the included articles, the following characteristics were extracted about the datasets: name of the datasets used, whether it was open-access or required credentials, the type of data used (e.g. tabular, images, etc.), the originating country of the dataset, and whether or not the authors of the paper created the dataset cited. Further, author nationality was extracted and was based on the country of the affiliated institution of the author. This was used to determine the presence or absence of a Latin American author amongst the group.

## Supporting information

**S1 PRISMA Checklist. PRISMA 2009 Checklist.**
(PDF)

**S1 File. Search Strategy and Criteria.**
(DOCX)

## Author Contributions

**Conceptualization:** David Restrepo, Luis Filipe Nakayama.

**Data curation:** David Restrepo, Justin Quion, Luis Filipe Nakayama.

**Formal analysis:** David Restrepo, Justin Quion, Luis Filipe Nakayama.

**Investigation:** David Restrepo, Constanza Vásquez-Venegas, Luis Filipe Nakayama.

**Methodology:** David Restrepo, Justin Quion, Luis Filipe Nakayama.

**Software:** David Restrepo.

**Validation:** Luis Filipe Nakayama.

**Visualization:** David Restrepo.

**Writing – original draft:** David Restrepo, Justin Quion, Luis Filipe Nakayama.

**Writing – review & editing:** David Restrepo, Justin Quion, Constanza Vásquez-Venegas, Cleva Villanueva, Leo Anthony Celi, Luis Filipe Nakayama.

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
