## [Decision Letter · Decision Letter 0]

21 Aug 2023

PDIG-D-23-00259

A Scoping Review of The Landscape of Health-Related Open Datasets in Latin America

PLOS Digital Health

Dear Dr. Restrepo,

Thank you for submitting your manuscript to PLOS Digital Health. After careful consideration, we feel that it has merit but does not fully meet PLOS Digital Health's publication criteria as it currently stands. Therefore, we invite you to submit a revised version of the manuscript that addresses the points raised during the review process.

Please submit your revised manuscript within 30 days Sep 20 2023 11:59PM. If you will need more time than this to complete your revisions, please reply to this message or contact the journal office at digitalhealth@plos.org. Please include the following items when submitting your revised manuscript:

We look forward to receiving your revised manuscript.

Kind regards,

Crina Grosan

Academic Editor

PLOS Digital Health

Journal Requirements:

2. Please send a completed 'Competing Interests' statement, including any COIs declared by your co-authors. If you have no competing interests to declare, please state "The authors have declared that no competing interests exist". Otherwise please declare all competing interests beginning with twhe statement "I have read the journal's policy and the authors of this manuscript have the following competing interests:"

3. Please ensure that Funding Information and Financial Disclosure Statement are matched.

4. In the Funding Information you indicated that no funding was received. Please revise the Funding Information field to reflect funding received.

5. Some material included in your submission may be copyrighted. According to PLOS’s copyright policy, authors who use figures or other material (e.g., graphics, clipart, maps) from another author or copyright holder must demonstrate or obtain permission to publish this material under the Creative Commons Attribution 4.0 International (CC BY 4.0) License used by PLOS journals. Please closely review the details of PLOS’s copyright requirements here: PLOS Licenses and Copyright. If you need to request permissions from a copyright holder, you may use PLOS's Copyright Content Permission form.

Potential Copyright Issues:

Figure 2: please (a) provide a direct link to the base layer of the map (i.e., the country or region border shape) and ensure this is also included in the figure legend; and (b) provide a link to the terms of use / license information for the base layer image or shapefile. We cannot publish proprietary or copyrighted maps (e.g. Google Maps, Mapquest) and the terms of use for your map base layer must be compatible with our CC-BY 4.0 license. 

"

Additional Editor Comments (if provided):

Reviewers' comments:

Reviewer's Responses to Questions

**Comments to the Author**

1. Does this manuscript meet PLOS Digital Health’s publication criteria? Is the manuscript technically sound, and do the data support the conclusions? The manuscript must describe methodologically and ethically rigorous research with conclusions that are appropriately drawn based on the data presented.

Reviewer #1: Yes

Reviewer #2: Yes

2. Has the statistical analysis been performed appropriately and rigorously?

Reviewer #1: Yes

Reviewer #2: N/A

3. Have the authors made all data underlying the findings in their manuscript fully available (please refer to the Data Availability Statement at the start of the manuscript PDF file)?

Reviewer #1: Yes

Reviewer #2: Yes

4. Is the manuscript presented in an intelligible fashion and written in standard English?

Reviewer #1: Yes

Reviewer #2: Yes

5. Review Comments to the Author

Reviewer #1: Dear Authors,

This is an excellent review paper, which I enjoyed reading, and it demonstrates the significance of open health data in Latin America. I would like to note that the attached review report can be discussed further if necessary. I tried to be as constructive as possible and offer you a wide range of possible actions. I am available via email or editorial at any time.

Congratulations and many thanks to the community for such an extensive review and list of suggestions.

Note: To avoid any confusion, the attached review mentions "Major considerations", but this is a minor revision overall, far from a major one. However, I believe this will have a significant impact on the research community this review is aimed at, hence the "Major recommendations" title.

Best wishes,

Reviewer #2: The article by Restrepo and colleagues is a review article presenting the status of health-related open datasets in Latin America. Authors identify existing datasets and examine their format, sharing framework, authors’ institution etc. This was an interesting work, treating the timely topic of diversity in open data sharing. I found the article well written and engaging. I have some suggestions for the authors listed below.

1. The review is based on article searches, but what about datasets that may be openly shared but not part of a paper that your search may have yielded? Could you discuss this as a potential limitation?

2. Section “Authors”: “In 120 (85.1%) of the articles, at least one Latin American author was included. It should be noted nationality was based on the location of the affiliated institution” By reading this it seems that what was evaluated was not the nationality, but the country where authors work. If this is indeed the case, could you consider rephrasing to “at least one author was working in a Latin American institution” and also rephrasing the term “nationality” as it is possible that other nationalities work in Latin American institutions. The same applies to the section “Conclusions” page 11. 

3. Discussion: “despite the combined scope, some papers and databases may have been omitted, although we argue that databases should be “findable” if they are to be useful”. While I agree with this argument, it is technically possible that the curators of some datasets did not publish a paper, and may expect their data to be findable in data repositories, for example the OSF, or domain-specific repositories. It may be worth adding a note about the possibility of having findable datasets but only in data repositories and not as a publication.

4. Section: “Support Funding and Infrastructure”. This is an excellent point. Without dedicated funding and infrastructure it is practically impossible to have open access data. I believe that in several countries funding agencies mandate that any research that is funded by them needs to openly release the resulting data. I am not aware of whether this is the case in any Latin American countries, but it may be worth to mention as a possible future direction of action.

5. “Other possibilities include establishing a rigorous de-identification process and implementing data governance practices” : In several if not most cases of medical data rigorous de-identification is not possible, as personal health data can act as a fingerprint. Could you elaborate on what needs to be done in cases where rigorous de-identification is not possible? Also, a full stop is missing from the sentence.

6. Section: “Foster data sharing culture”: “Health institutions should be made aware that sharing data is important and represents a benefit and not a risk.” Although I agree that sharing data generated in health institutions is a benefit, there are also risks associated with that, for example related to privacy, identification, etc. Could you elaborate on that?

6. PLOS authors have the option to publish the peer review history of their article (what does this mean?). If published, this will include your full peer review and any attached files.

**Do you want your identity to be public for this peer review?** For information about this choice, including consent withdrawal, please see our Privacy Policy.

Reviewer #1: Yes: Simon Gilbert Provost

Reviewer #2: No

---

## [Decision Letter · Decision Letter 1]

16 Sep 2023

A Scoping Review of The Landscape of Health-Related Open Datasets in Latin America

PDIG-D-23-00259R1

Dear Eng. Restrepo,

We are pleased to inform you that your manuscript 'A Scoping Review of The Landscape of Health-Related Open Datasets in Latin America' has been provisionally accepted for publication in PLOS Digital Health.

Best regards,

Crina Grosan

Academic Editor

PLOS Digital Health

Reviewer Comments (if any, and for reference):

Reviewer's Responses to Questions

**Comments to the Author**

1. If the authors have adequately addressed your comments raised in a previous round of review and you feel that this manuscript is now acceptable for publication, you may indicate that here to bypass the “Comments to the Author” section, enter your conflict of interest statement in the “Confidential to Editor” section, and submit your "Accept" recommendation.

Reviewer #1: All comments have been addressed

Reviewer #2: All comments have been addressed

2. Does this manuscript meet PLOS Digital Health’s publication criteria? Is the manuscript technically sound, and do the data support the conclusions? The manuscript must describe methodologically and ethically rigorous research with conclusions that are appropriately drawn based on the data presented.

Reviewer #1: Yes

Reviewer #2: (No Response)

3. Has the statistical analysis been performed appropriately and rigorously?

Reviewer #1: Yes

Reviewer #2: (No Response)

4. Have the authors made all data underlying the findings in their manuscript fully available (please refer to the Data Availability Statement at the start of the manuscript PDF file)?

Reviewer #1: Yes

Reviewer #2: (No Response)

5. Is the manuscript presented in an intelligible fashion and written in standard English?

Reviewer #1: Yes

Reviewer #2: (No Response)

6. Review Comments to the Author

Reviewer #1: Dear Authors,

I concur with the latest revision. The comments were meticulously taken into consideration and analysed as effectively as possible. Regarding your comment about the Auto-ML's unaffordable technique for accelerating research in these regions, I concur and trust your experience; hopefully, something will improve in the future for these regions.

In the meantime, I am accepting the paper and offer my congratulations to the entire team that collaborated together !

Best wishes,

Reviewer #2: (No Response)

7. PLOS authors have the option to publish the peer review history of their article (what does this mean?). If published, this will include your full peer review and any attached files.

**Do you want your identity to be public for this peer review?** For information about this choice, including consent withdrawal, please see our Privacy Policy.

Reviewer #1: **Yes: **Simon Gilbert Provost

Reviewer #2: No
